# Characterization of 22q12 Microdeletions Causing Position Effect in Rare NF2 Patients with Complex Phenotypes

**DOI:** 10.3390/ijms231710017

**Published:** 2022-09-02

**Authors:** Viviana Tritto, Marica Eoli, Rosina Paterra, Serena Redaelli, Marco Moscatelli, Francesco Rusconi, Paola Riva

**Affiliations:** 1Dipartimento di Biotecnologie Mediche e Medicina Traslazionale, Università degli Studi di Milano, 20054 Segrate, Italy; 2Unità di Neuro-Oncologia Molecolare, Fondazione IRCCS, Istituto Neurologico Carlo Besta, 20133 Milan, Italy; 3Dipartimento di Medicina e Chirurgia, University of Milano-Bicocca, 20900 Monza, Italy; 4Unità di Neuroradiologia, Fondazione IRCCS, Istituto Neurologico Carlo Besta, 20133 Milan, Italy

**Keywords:** NF2, 22q12 microdeletion, position effect, haploinsufficiency, non-allelic homologous recombination, genotype-phenotype correlation

## Abstract

Neurofibromatosis type 2 is an autosomal dominant tumor-prone disorder mainly caused by *NF2* point mutations or intragenic deletions. Few individuals with a complex phenotype and 22q12 microdeletions have been described. The 22q12 microdeletions’ pathogenic effects at the genetic and epigenetic levels are currently unknown. We here report on 22q12 microdeletions’ characterization in three NF2 patients with different phenotype complexities. A possible effect of the position was investigated by in silico analysis of 22q12 topologically associated domains (TADs) and regulatory elements, and by expression analysis of 12 genes flanking patients’ deletions. A 147 Kb microdeletion was identified in the patient with the mildest phenotype, while two large deletions of 561 Kb and 1.8 Mb were found in the other two patients, showing a more severe symptomatology. The last two patients displayed intellectual disability, possibly related to *AP1B1* gene deletion. The microdeletions change from one to five TADs, and the 22q12 chromatin regulatory landscape, according to the altered expression levels of four deletion-flanking genes, including *PIK3IP1*, are likely associated with an early ischemic event occurring in the patient with the largest deletion. Our results suggest that the identification of the deletion extent can provide prognostic markers, predictive of NF2 phenotypes, and potential therapeutic targets, thus overall improving patient management.

## 1. Introduction

Neurofibromatosis type 2 (NF2; MIM # 101000) is a rare autosomal dominant condition caused by mutations in the *NF2* gene (Neurofibromin 2; MIM # 607379), with an incidence of 1 in 33,000 and a prevalence of 1 in 56,000 [1]. In 95% of patients, this disorder is associated with the development of bilateral vestibular schwannomas (VSs), also known as acoustic neuromas, affecting balance and hearing nerves, leading to deafness [2]. NF2 patients can also develop other tumors, including schwannomas of the cranial, spinal, peripheral, and cutaneous nerves, but also meningiomas and ependymomas [3]. Other clinical manifestations include ophthalmic signs, such as decreased visual acuity, retinal hamartoma, and cataracts [4], along with dermatological features, such as dermal plaque-like lesions, subcutaneous nodular tumors, and café-au-lait spots [3,5]. In most cases, NF2 presents at adult age with hearing loss related to VSs. However, concerning childhood-onset NF2, no vestibular manifestations such as neuropathy or eye axis deviation occur, underlining the need to focus more closely on early-onset symptomatic traits in children, ultimately allowing for timely diagnostic assessments. Molecular diagnosis is based on the identification of heterozygous pathogenic variants in the *NF2* gene, located on 22q12.2, by molecular genetic testing.

The *NF2* gene encodes Moesin–Ezrin–Radixin-Like (Merlin), a 595-amino-acid protein that belongs to the ERM (ezrin, radixin, and moesin) protein family, known to mediate the interaction of cytoskeletal components with cell membrane proteins, involved in the control of cytoskeletal dynamics and ion transport [6]. This protein also represents a critical modulator of the contact-dependent inhibition of cell proliferation, regulating the Hippo/SWH (Sav/Wts/Hpo) signaling pathway, which plays a pivotal role in tumor suppression by restricting proliferation and promoting apoptosis [7]. Merlin also exerts a negative control effect on other proliferative signaling pathways, including RAS-MAPK, FAK/Src, PI3K/AKT/mTOR, Rac, and Wnt/β-catenin pathways [8,9]. Therefore, *NF2* is a “tumor suppressor” gene and, as such, the related syndrome is caused by loss-of-function mutations of this gene. About half of NF2 patients do not have any family history showing de novo *NF2* mutations, and up to 60% of them may display a mosaic condition due to post-zygotic mutations, as well as a mild phenotype [10,11]. Most *NF2* alterations consist of truncating mutations, such as nonsense variants and frameshift deletions/insertions, but splice variants, missense mutations, and in-frame deletions/insertions were also detected. Single and multiple exons, or whole *NF2* gene deletions, can be found in about 15% of affected individuals, while only rare cases of larger-scale chromosomal rearrangements involving the chromosome 22q12, including ring chromosome, translocations, and microdeletions spanning the whole *NF2* sequence and flanking genes, have been reported [12,13,14,15].

Several studies have established clear genotype–phenotype correlations in NF2 [16,17]. Interestingly, patients carrying truncating mutations present a more severe phenotype compared to those who show missense mutations, splice site mutations, or large deletions, which are all frequently associated with mild NF2 [11,18,19]. In particular, this latter group of patients shows delayed disease onset and diagnosis, developing fewer meningiomas, spinal tumors, and peripheral nerve tumors. The specific position of the mutations correlates with disease severity: splice site mutations located in the first five exons and truncating mutations in exons 2–13 of *NF2* gene, causing premature termination of the protein, have been associated with severe phenotypes, while splice site mutations in exons 11–15 and 3’ truncating mutations (exons 14–15) entail decreased tumor incidence and lower mortality [17,20]. Large deletions, including the promoter or the first exon of the *NF2* gene, were found in patients with a milder phenotype than in clinical cases with a normal asset of this part of the gene [16].

Since 1993, few individuals with 22q12 large deletions or microdeletions, including *NF2* and flanking genes, have been described [15,21,22,23,24,25,26,27,28,29]. However, due to the large temporal frame, most patients have not been fully described, and the actual care pathway has not always been established. Furthermore, large deletions have not been precisely characterized. In addition, Baser, Hexter, and Selvanathan, who compared the clinical presentations and overall survival of patients with mutations that are predicted to produce truncated proteins (nonsense/frameshift mutations) to those with mutations that result in loss of protein expression (large deletions), confirm that nonsense/frameshift mutations are associated with more severe NF2 symptoms. However, all data refer to the same data set (population-based United Kingdom NF2 registry), and microdeletion patients and cases with other large deletions have been grouped and analyzed together.

To allow patient stratification according to disease severity, the UK NF2 Reference Group established a Genetic Severity Score (GSS) based on the typology of the *NF2* germline variant observed [11]. The GSS was validated in a Spanish NF2 cohort and a recent revision of the GSS, named the Functional Genetic Severity Score (FGSS), was proposed [30]. The main dissimilarities between GSS and FGSS regard the classification of large deletions and splicing variants. Nevertheless, large deletions in *NF2* patients are mostly inadequately analyzed, and deletion extent, along with inherent gene content, remains elusive. Usually, NF2 patients are assayed for the presence of *NF2* gene point mutation by sequencing approaches. *NF2* gene deletions are commonly assessed by MLPA analysis (Multiplex Ligation-Dependent Probe Amplification Analysis), disregarding, as said above, the precise characterization of the deletion extent, which critically limits genotype–phenotype correlations. More recently, the development of NGS techniques and dedicated bioinformatic pipelines has allowed us to identify both point mutations and copy number changes and thus to better address breakpoint isolation. Unfortunately, the use of this technique is not yet familiar enough to be widely applied in the diagnostic laboratory.

We here describe three patients carrying rare 22q12 microdeletions. Two out of the three patients with a presumed de novo deletion, of 1.86 Mb and 561 Kb, are characterized by a more severe phenotype than that observed in the third patient, who carried a smaller inherited deletion of 146.8 Kb. Breakpoint characterization allowed us to establish the microdeletion gene content, thus approaching genotype–phenotype correlations. Furthermore, an expression analysis of microdeletion flanking genes revealed, for the first time in NF2 microdeletion patients, the occurrence of an effect of position. The obtained results could provide useful information that would allow an earlier diagnosis and help establish a personalized NF2 prognosis.

## 2. Results

### 2.1. Clinical Description

Here, we describe three patients carrying rare 22q12 microdeletions. The patients’ clinical features are summarized in Table 1.

#### 2.1.1. Patient 246

A young female, living in a low-income country, was referred with dizziness and gait disturbance at the age of 23 years. Her father, paternal aunt, and grandfather suffered from hearing loss due to VS. The father’s and aunt’s clinical files were checked; it was found that hearing impairment, dizziness, and gait disturbance occurred after puberty, and both had bilateral VS and no meningioma. They died at 42 and 38, respectively, due to intracranial hypertension, and surgery was not performed.

This patient underwent a brain MRI with gadolinium, showing the presence of bilateral VSs, and Neurofibromatosis type 2 was diagnosed. At the age of 27, she was treated with gamma-knife on the right VS, because of progressive hearing impairment. A year later, when the brain MRI revealed a volumetric increase in the left VS, a gamma-knife was applied to the left VS as well. No other lesions were observed by spinal MRI or cerebral magnetic resonance angiography of the head. At present, physical examinations show café-au-lait spots on the glutes, gait disturbance, and hearing impairment that is more relevant on the right side for high frequencies. Genetic testing has revealed a large deletion of the *NF2* region.

#### 2.1.2. Patient 366

The family history was unremarkable. At 7 years old, learning and hyperactivity disorders were observed, and a global developmental delay was then diagnosed. At 13 years old, due to persistent enuresis, spinal MRI with gadolinium was ordered, showing a large enhancing lesion at the S1 level. NF2 was suspected, and brain MRI with gadolinium revealed bilateral VS. At the age of 30, due to progressive hearing loss with increased VS volume and mass effect with distortion of pons profiles, a ventricular peritoneal shunt was performed along with gross total removal of the right VS, followed by gamma-knife treatment. The patient developed complete right-side hearing loss. An eye examination showed bilateral nuclear cataracts. The cerebral magnetic resonance angiography of the head was normal. A new spinal MRI with gadolinium showed suspected schwannomas at C2-C4, C7-T1, and T10-S1. At present, physical and neurological examinations show two nuanced café-au-lait spots and small schwannomas on the chest, while mild intellectual disability was scored as 22 by the Mini-Mental State Exam (MMSE) and 20 by the Wechsler Adult Intelligence Scale. Gait disturbance, nystagmus, right VII palsy, and hearing impairment completed the diagnosis. Genetic testing revealed a large deletion of the *NF2* region. The deletion was not present in the patient’s mother, and the father was not available for analysis, even though, at 55 years old, he did not have any NF2 symptoms or signs.

#### 2.1.3. Patient 160

A 3-year-old boy presented with sudden left lower and upper limb hemiparesis and dysarthria. A brain MRI showed right pons hyperintensity without enhancement in T2 weight images, as well as hypoplasia of the corpus callosum, and an ischemic stroke was suspected. The patient’s condition improved without full recovery in one month with a mild residual left hemiparesis. No cardiovascular risk factor was present, and cardiac, vascular, and the laboratory diagnosis did not suggest extracranial stroke. In addition, the patient’s education suffered considerably, and a global developmental delay was diagnosed. At the age of 7, he presented ptosis and III nerve palsy of the left eye. At the age of 17, physical examination revealed the presence of a subcutaneous lump on the left ankle, which was removed; the histological diagnosis was schwannoma. At the age of 20, progressive hearing impairment and tongue swelling were observed. The lump was removed with a diagnosis of schwannoma. At age 22 years, the patient underwent a new brain MRI with gadolinium, showing bilateral VSs with a mass effect and the distortion of pons profiles and the IV ventricle, marked hypoplasia of the corpus callosum associated with widely spaced lateral ventricles, and small schwannomas on the left III cranial nerve (Appendix A). One year later, NF2 was diagnosed. Spinal MRI with gadolinium revealed enhanced focal lesions suggestive of schwannomas at the C3 left root, at the C6 bilateral roots, at the D9 left root, at the S1 left root, and along the cauda roots. Cerebral magnetic resonance angiography, including vertebro-basilary arteries of the head, was normal. Eye examination showed left III cranial nerve palsy. Audiometric examination showed severe hearing impairment that was more relevant on the left side, and a hearing aid was needed. Dermatological features consisted of a café-au-lait spot on the right armpit, two schwannomas at the right hand, and two NF2 plaques on the right chest. The MMSE was 24, and the Wechsler Adult Intelligence Score was 21. At the age of 23, therapy with bevacizumab 5 mg/kg every two weeks was started and suspended 14 months later due to hypertension and persistent proteinuria. Genetic testing revealed a large deletion of the *NF2* region. The deletion was not present in the patient’s mother and father.

### 2.2. Identification of 22q12 Microdeletions by MLPA

A *NF2* mutation screening was performed on the genomic DNA extracted from the peripheral blood of patient 246, 366, and 160 using an NGS panel, including *NF2*, *LZTR1*, and *SMARCB1* genes on Ion Torrent, and no mutation on those genes was found. Thus, the MLPA analysis was performed to detect the possible presence of a deletion involving the *NF2* gene. A deletion in the 22q12 chromosomal region was diagnosed in each of the three patients.

In patient 246, the proximal breakpoint (BP) was included within a 5,775,890 bp region, between the last proximal not in-loss probe (08294-L20837 of P044 NF2 C1), mapped in exon 8 of the *SMARCB1* gene, and the first in-loss probe (02580-L02042), localized within *NIPSNAP1* exon 10. The distal BP was delimited in a 3246 bp region by the last in-loss probe (01577-L01149) and the first distal not in-loss probe (22440-L31615), mapping in exons 14 and 15 of the *NF2* gene, respectively (Appendix A).

Patients 366 and 160 showed the same MLPA profile. The genomic region containing the proximal BP corresponded to that identified in patient 246, while the distal BP was localized within a 3,655,027 bp region, between the last in-loss probe (03317-L31857), in the *CABP7* exon 4, and the first distal not in-loss probe (12460-L13461), in the *LARGE1* exon 9 (Appendix A).

### 2.3. Definition of 22q12 Microdeletions Extent by gDNA-qPCR and aCGH

To narrow down the region containing the proximal BP of patient 246’s microdeletion, we performed two consecutive qPCR assays on gDNA extracted from the patient’s peripheral blood (Appendix A), whereas the distal BP region was not further narrowed down as it was already resolved by MLPA. The first qPCR assay showed that all four regions analyzed, mapping along the entire interval containing the proximal BP identified by MLPA analysis, were double copied in the genome of patient 246 compared to intron 5 of the *NF2* gene, chosen as a single copy control. The 565,532 bp region thus identified was further narrowed to 110,606 bp by a second qPCR assay. According to the qPCR data, the proximal BP was located between the downstream region of the *RFPL1* gene and the *NIPSNAP1* exon 10 (Figure 1).

To better define the extent of the microdeletions for patients 366 and 160, an aCGH analysis was performed on the gDNA extracted from the patients’ peripheral blood (Appendix A). According to the aCGH data, patient 366 showed a 22q12 microdeletion ranging between 515,173 and 574,371 bp. The proximal BP was within an interval region of 32,405 bp, while the distal BP was included in the region of 26,795 bp (Figure 1).

The extent of patient 160’s microdeletion, identified by aCGH analysis, ranged from 1,836,171 to 1,897,805 bp. The proximal BP fell into a region of 29,669 bp, while the region containing the distal BP was around 31,967 bp (Figure 1).

### 2.4. Identification of Microdeletions’ BPs and Deletion Gene Content

To identify the exact genomic location of the proximal and distal BPs, long-range PCR assays were carried out on the gDNA extracted from the patients’ peripheral blood. Because the 22q12 region is very rich in repeats, we hypothesized that they were involved in the occurrence of the unequal homologous recombination mechanism, causing the deletions under study. We searched for repeated motifs showing at least 75% homology after in silico analysis of the BP regions. For each patient, a set of forward and reverse oligonucleotides was designed, respectively, in the proximal and distal BP regions, upstream of all repeated sequences showing the established homology degree (Appendix A). The junction fragments of the microdeletions of patients 246, 366, and 160 were isolated by specific amplification products of about 700 bp, 1.3 kb, and 300 bp, respectively, and directly sequenced (Appendix A).

For patient 246, the proximal BP was located in intron 7 of the *THOC5* gene, within a sequence of 8 bp (chr22:29,928,889-29,928,896, GRCh37/hg19), included in the region with the distal BP (chr22:30,075,777-30,075,784, GRCh37/hg19), mapped in intron 14 of *NF2* (Figure 2A). The proximal and distal BPs fell into two sequences of about 300 bp belonging to the L1ME3 and MLT1A0 repeat elements, respectively, showing a sequence homology of about 80%. According to the BPs’ positions, patient 246’s microdeletion extended for 146,888 bp and caused the total loss of the *NIPSNAP1* gene and the partial deletion of the *THOC5* and *NF2* genes (Appendix A). As the orientation of the *THOC5* and *NF2* genes was towards the minus and plus strands, respectively, both genes lost the 5’ region, including promoters. Furthermore, no ORFs have been predicted. Therefore, the generation of an unfunctional chimeric gene was expected.

In patient 366, the proximal BP fell in an intergenic region upstream of the *EMID1* gene, within a sequence of 33 bp (chr22:29,594,735-29,594,767, GRCh37/hg19) identical to the region including the distal BP (chr22:30,155,876-30,155,908, GRCh37/hg19), into intron 1 of *ZMAT5* (Figure 2B). The rearrangement led to the removal of most of the *ZMAT5* coding region. The proximal and distal BPs were mapped in two repeated sequences, Tigger2b_Pri and AluSx, respectively. The two regions of about 180 bp showed 83% homology. Based on the BPs’ localizations, patient 366’s microdeletion included 561,141 bp, counting 13 protein-coding genes (Appendix A).

In patient 160, the proximal BP mapped in an intergenic region upstream of the *ZNRF3* gene, within a 15 bp sequence (chr22:29,277,234-29,277,248, GRCh37/hg19) in common with the region including the distal BP (chr22:31,143,939-31,143,953, GRCh37/hg19), located in intron 2 of *OSBP2* (Figure 2C). This deletion led to the loss of *OSBP2*’s first two exons and its promoter. The proximal and distal BPs fell into two sequences of about 220 bp, which had 83% homology, belonging to the L1MB7 and L2b repeat elements, respectively. According to the BPs’ positions, patient 160’s microdeletions involved 38 protein-coding genes, for an extension of 1,866,705 bp (Appendix A).

Genotype–phenotype correlations were evaluated, considering the possible association between the haploinsufficiency of genes included in the 22q12 microdeletions and specific clinical signs shown by our patients. The haploinsufficiency was estimated based on the gene’s relative probability of being loss-of-function-intolerant (pLI), a score ranging from 0 to 1, wherein a value equal to 1, typical of haploinsufficient genes, indicates the maximum degree of intolerance toward the specific loss of function. The microdeletion of the patient with the mildest phenotype, patient 246, included only the *NF2* gene with a high pLI, equal to 1.00. Patients 366 and 160, both affected by more severe symptoms, including mental retardation, unusual in the NF2 phenotypic spectrum, showed three (*EWSR1*, *AP1B1*, *NF2*) and six (*ZNRF3*, *EWSR1*, *AP1B1*, *NF2*, *MTMR3*, and *SF3A1*) genes with pLI greater than 0.9, respectively. Among these genes, *AP1B1* has previously been reported to be associated with MEDNIK-like syndrome (intellectual disability, enteropathy, deafness (sensorineural), neuropathy, ichthyosis, and keratodermia) [31], which shows an autosomal recessive inheritance. Therefore, quantitative RT-PCR on the RNA extracted from the peripheral blood of patients 366 and 160 and controls was carried out to evaluate the expression of the second *AP1B1* allele. The gene expression analysis showed a 50% decreased *AP1B1* expression in the two patients (Figure 3). Furthermore, to verify the presence of loss-of-function mutations in the *AP1B1* non-deleted allele, possibly unmasking a pseudodominance effect, sequencing analysis of the *AP1B1* gene transcript has been carried out. We identified in both patients two synonymous mutations, c.204A>G (NM_001127.4) and c.2349G>A (NM_001127.4), classified as benign in ClinVar. The *AP1B1* screening results are consistent with the presence of a functional allele.

### 2.5. Position Effect on 22q12 Microdeletions Flanking Genes

In order to evaluate possible position effects that can alter the expression of 22q12 microdeletions-flanking genes, we performed gene expression analysis of candidate genes by quantitative RT-PCR on the RNA extracted from peripheral blood, comparing our microdeletion-carrying *NF2* patients to ten healthy controls (seven females and three males, with an average age of 39 years) and three patients with *NF2* intragenic mutations (two males and one female, average age 37 years). This control was implemented to ascertain whether possible modifications in the regulation of microdeletion-flanking genes were caused by microdeletion-dependent alterations of topological DNA domains, instead of altered pathways involving the *NF2*’s direct or the indirect modulation of gene expression. For the qPCR assays, we selected 12 ubiquitously expressed genes with an expression level greater than 2 TPM (transcripts per million) in the whole blood, mapping on the 22q12 chromosome region within the TADs in which the BPs of the patients’ deletions fell (Figure 4A). Compared to WT controls and NF2 patients with intragenic mutations, patient 246 showed a 75% reduced transcript expression of the *AP1B1* gene, mapped in a centromeric region to the deletion, as well as increased expression of the *MTMR3* and *LIMK2* genes, located at the telomeric position. Patient 366 showed hypo-expression of *MTMR3*. *LIMK2* and *PIK3IP1,* also located downstream of the telomeric breakpoint, were found to be overexpressed in patient 160. The analysis of three independent biological samples of the genes found to be dysregulated demonstrated statistically significant differences for all genes only in patients 246 and 160. Patient 246 showed the statistically dysregulated expression of *AP1B1* and *MTMR3* through the comparison with healthy controls, and of *LIMK2*, compared to both wild type controls and NF2 patients carrying intragenic mutations. *LIMK2* and *PIK3IP1* were statistically hyper-expressed in patient 160, compared to NF2 patients with intragenic mutations (Appendix A; Figure 4B). According to the identified gene expression profiles, our results indicate a clear position effect of 22q12 microdeletions in patients’ peripheral blood. It is notable that, when comparing microdeletion-carrying NF2 patients to healthy controls, but not to intragenic mutations-carrying NF2 patients, the reduced mRNA expressions of two centromeric genes, namely, *HSCB* and *XBP1*, and of three telomeric genes (*ZMAT5*, *UQCR10*, and *ASCC2*), were also detected. The altered expressions of these five transcripts proved similar in the two categories of NF2 patients compared to healthy controls, suggesting that mechanisms other than the position effect, and probably directly or indirectly caused by NF2-related signaling, are at play in the pathogenesis of this pathology.

To corroborate the hypothesis of the position effect due to 22q12 microdeletions, we investigated the possible alteration of the topological domain landscape within the chromosomal region, including the microdeletions. As shown in Figure 5, patient 246’s microdeletion was located within the boundaries of a single TAD, including the *AP1B1* and *MTMR3* genes showing a deregulated expression. Patient 366’s microdeletion partially involved two TADs, the same TAD as in the previous patient and the upstream one, entailing the loss of a boundary between them. Patient 160’s microdeletion affected five TADs, two of which were only partially deleted, while three were totally removed, together with four TAD boundaries, one of which was to a considerable extent. Moreover, the two genes with statistically significant alterations of expression in patient 160 mapped distally to the telomeric BP, within the partially deleted TAD.

We carried out a prediction of the insulators and searched for enhancers present in the region of interest. We found two enhancers, both located outside the patients’ microdeletions—one centromeric and one telomeric to the deletions. Concerning the insulators, within this genomic region, there were 35 (Figure 5). All of these lay outside of patient 246’s microdeletion. Four were lost through patient 366’s microdeletion, and eighteen were lost through patient 160’s microdeletion (Appendix A). The obtained data indicate the TAD modification and consequent alterations of enhancers and insulators, explaining the position effect, which deregulates the expressions of the described microdeletions flanking the genes.

## 3. Discussion

Constitutional *NF2* deletion affects 11% to 24% of NF2 patients [26,32]. In most cases, deletions involve multiple exons of the *NF2* gene, only rarely encompassing larger genomic regions including multiple genes. Few large microdeletions have been characterized before. A mild NF2 phenotype has been reported in a case with the NF2 deletion extending to the EWS gene, which is 350 kb centromeric to NF2 [21], and in two other individuals with deletions around the NF2 locus, spanning approximately 700 kb [29,33]. Instead, Bruder described intellectual disability in a patient with a severe phenotype who displayed a 7.4 Mb deletion on chromosome 22 [13], and two other cases with 22 chromosome microdeletions of an undefined extent, intellectual disability, and early disease onset were published [34].

Smith, analyzing seven lymphocyte DNA samples from individuals with NF2 microdeletions, showed that the deletion sizes were different in every case, with one recurrent breakpoint identified in two samples. One of the seven deletions was 3.93 Mb long, and seemed to confer a severe phenotype. Based on this last finding and on the case reported by Bruder, the author suggested that a microdeletion extending to a 2.04 Mb region between 22:31,597,658 and 22:33,638,788 could confer a severe phenotype [28].

Consistently, the current guidelines for NF2 genetic screenings provide for the identification of specific *NF2* mutations, in some cases predicting the phenotype’s severity [11]. So far, the reported *NF2* microdeletions lack a precise characterization of their genomic coordinates as inferred from breakpoint cloning, only in a few cases allowing the identification of the deletion gene content. Here, we have reported on three NF2 patients carrying 22q12 microdeletions of different lengths. The deletions were precisely characterized, allowing us to establish the mechanisms underlying the deletions and the deletions’ gene content, and to investigate changes in the chromatin structure of deleted regions affecting gene expression.

The high homology of BP regions for each deletion, involving interspersed repetitive elements of 180, 220, and 315 nt, respectively, and all showing 70–80% homology, prompted us to hypothesize that these sequences may mediate non-allelic homologous recombination (NAHR) events at the basis of chromosomal deletions and duplications [35]. Notably, the q arm of chromosome 22 is rich in interspersed repetitive identical or nearly identical DNA sequences, scattered throughout the genome as a result of duplication, transposition, or retrotransposition events. In particular, in the 22q11 region, low copy repeats (LCRs) have been directly implicated in chromosomal rearrangements causing DiGeorge, velocardiofacial, and conotruncal anomaly face syndromes (DGS/VCFS/CAFS); cat eye syndrome (CES); and the supernumerary der(22)t(11;22) syndrome [36,37], generating deletions to a typical extent [38]. Differently, the 22q12 *NF2* region (chr22:28,250,000–32,050,000, GRCh37/hg19) is characterized by several families of repeated sequences that can be potentially implicated in recombination events generating deletions of different locations and sizes. What we observed in the microdeletion BPs of the NF2 patients under study confirms that repeated elements with a reciprocal homology greater than 80% are involved in rearrangements leading to *NF2* large deletions. The deletion BPs of patient 246 were located centromerically to the sequence L1ME3, which belongs to the class of LINE (long interspersed nuclear elements), and telomerically to MLT1A0, which is an LTR (long terminal repeat elements). The deletion BPs of patient 366 involve, centromerically, the Tigger2b_Pri sequence (a DNA transposon), and telomerically an AluSx element, belonging to SINE (short interspersed nuclear elements). The deletion of patient 160 was caused by a recombination between two LINE sequences, namely, the proximal L1MB7 and the distal L2b. The absence of LCRs and the presence of several repetitive elements make the 22q12 region prone to chromosomal rearrangements, leading to a variety of genetic lesions, and making the recurrence of recombination sites unlikely. The absence of recombination hotspots limits the implementation of specific genetic tests aimed at defining *NF2*’s deletion extent. The detection of *NF2* gene deletion is not sufficient per se to allow genotype–phenotype correlation. In fact, *NF2* gene deletion is not predictive of clinical phenotype severity, as occurs with NF1 microdeletion syndrome, in which large NF1 deletions, mostly of type 1, are associated with severe clinical phenotypes [39,40].

Disease onset can be considered, to some extent, as an accurate predictor of the natural history of the disease. All three patients showed early disease onset, during the first years of childhood for the two patients with larger deletions. The UK NF2 Genetic Severity Score [11] and the Functional Genetic Severity Score [30] represent useful tools to predict disease course. However, as far as large deletions are concerned, they only distinguish large deletions including the *NF2* promoter or exon 1 (group 2A) from those that do not (group 2B). While both groups are associated with a mild phenotype, it appears that at least in some cases, 22q12 microdeletions, including *NF2* and flanking genes, can be related to a more severe phenotype [24]. Smith and colleagues reported on two large NF2 deletions in two patients with a severe phenotype possibly associated with a putative modifier gene located in a 2 Mb region telomeric to *NF2* [28].

Given that the deletion of the *NF2* gene is associated with a mild disease phenotype, we inferred that NF2 patients carrying microdeletions, including additional genes besides *NF2*, show a more severe phenotype because of the involvement of causative mechanisms, such as haploinsufficiency of more than one gene differently deleted in NF2 deletions with different localizations. This hypothesis is consistent with the evidence that the patient with the smallest deletion (patient 246), of about 147 Kb, has a milder phenotype than that observed in patients 366 and 160, who instead carry larger deletions of about 561 Kb and 1.8 Mb, respectively.

The precise identification of the deletions’ BPs of the three patients allowed us to study their gene contents and evaluate the pLI scores. Patient 246 showed a classic NF2 phenotype, generally associated with the heterozygous *NF2* mutation and consistent with the gene content of his deletion, which includes, in addition to *NF2*, only two other genes, *THOC5* and *NIPSNAP1*, both characterized by a low pLI. Patients 366 and 160 both showed intellectual disability, which is unusual in the NF2 phenotypic spectrum. According to Halliday, developmental delay is observed in about 3% of NF2 children—only two NF2 cases with 22 chromosome microdeletions of undefined extents, one displaying ring chromosome 22 and another with a mild delay and a small intragenic lesion, have previously been reported [34]. Bruder described intellectual disability in a patient with a severe phenotype who displayed a 7.4 Mb deletion on chromosome 22 [13]. Among the few genes with high pLI included in the deletions of patients 366 and 160, it is worth mentioning that the *AP1B1* gene, also included in the deletion described by Bruder, has previously been reported to be associated with MEDNIK-like syndrome [31]. However, this syndrome has an autosomal recessive inheritance. Because both patients showed synonymous variants in the *AP1B* gene transcript, classified as benign in ClinVar, we suggest that the function of the *AP1B1* non-deleted allele remains unaltered. The absence of loss-of-function mutations in both *AP1B1* alleles, besides the gene expression being reduced by half in both patients, could be suggestive of a mild expression of a MEDNIK-like syndrome phenotype, limited to intellectual disability.

Another pathomechanism that has been poorly investigated in microdeletion syndromes, but has possible consequences on clinical phenotype, is the position effect on the regulation of microdeletion-flanking genes’ expression [41,42,43]. In this work, we have provided, for the first time, findings suggesting a position effect associated with 22q12 chromosomal microdeletions in NF2 patients. To this extent, candidate gene expression was evaluated in patients’ blood, as other tissues were not available. Among these genes, we selected the ubiquitously expressed ones, as a proxy for expression deregulation in different tissues. The position effect was inferred, as gene expression analysis showed a trend toward dysregulation for the *AP1B1*, *MTMR3*, and *LIMK2* genes in patient 246, *MTMR3* in patient 366, and *LIMK2* and *PIK3IP1* in patient 160, although statistically significant expression variation was only demonstrated for all genes in patients 246 and 160. *AP1B1* (Adaptor Related Protein Complex 1 Subunit Beta 1) encodes the β subunit of heterotetrameric adaptor protein 1 (AP-1) complexes, which coordinate vesicles’ formation at the Golgi complex, the recruitment of clathrin to their membranes, and sorting signals [44]. *MTMR3* (Myotubularin Related Protein 3) encodes an inositol lipid 3-phosphatase belonging to the myotubularin family. Inositol lipids mediate several cellular functions, including proliferation, survival, and membrane trafficking [45]. *LIMK2* (LIM Domain Kinase 2) is a phosphoprotein that regulates actin cytoskeleton dynamics belonging to the Rho signaling pathway [46]. *PIK3IP1* (Phosphoinositide-3-Kinase Interacting Protein 1) is a transmembrane protein that inhibits the activation of the PI3K/Akt signaling pathway [47].

Non-vestibular disease onset with cranial nerve palsy is often described in children with NF2. However, our cases presented with atypical features, including sudden hemiparesis with dysarthria, learning disability, and neurodevelopment delay. Symptomatic pontine infarct without obvious reasons for stroke is rarely described in NF2 pediatric patients and young adults [34,48]. Interestingly, one of the few cases reported harbored a 22q microdeletion, including the *NF2* gene [34]. An early ischemic brain stem event should be considered a possible presenting feature of NF2 in childhood, and large deletions should be investigated. Notably, among the genes with altered expressions, *PIK3IP1* overexpression in patient 160 could be related to the ischemic event that affected the patient at the age of 3 years. Consistently, the overexpression of *PIK3IP1* is relevant in rat models of cerebral ischemia induced via MCAO (middle cerebral artery occlusion). Interestingly, it was suggested that the extent of the ischemic damage can be reduced through the use of miRNA to target *PIK3IP1* transcripts [47]. The expression dysregulation of the other genes is an informative indicator of the position effect of the studied deletions.

It is known that, at the basis of gene expression deregulation, an alteration of the microdeletion-flanking regulatory chromatin secondary to a modification of the three-dimensional structure of chromatin, organized in TADs, can be found [49]. We performed TAD mapping within the deleted regions, predicting regulatory elements’ disruptions that relate to the position effect on gene expression regulation. In particular, patient 246’s microdeletion involved neither insulators nor enhancers and boundaries between TADs, mapping within a single TAD. This alteration implies a position effect of an elusive nature, as indicated by *AP1B1* hypo-expression and *MTMR3* and *LIMK2* overexpression. On the contrary, the microdeletions in patients 366 and 160’, respectively, involved two and five TADs, causing the loss of several insulators. The deletion of these elements frequently leads to the alteration of enhancer–promoter interactions, with an effect on gene transcription [50]. Nevertheless, we detected alterations in gene expression profiles that were limited to a few genes in the blood. If, on the one hand, the housekeeping expression of such deregulated genes could suggest their possible alteration in other tissues as well, on the other hand, it is worth noting that in this region, TADs are not conserved among different tissues, which makes it difficult to draw predictions. A possibility is that gene expression in other tissues could even undergo more severe deregulation. As a matter of fact, poor tissue availability for gene expression analysis limits our further understanding of the impact related to the position effect. This study limitation could be addressed by generating iPSCs from patients’ cells, allowing for identifying druggable genes that could be used in future pharmacological interventions.

## 4. Materials and Methods

### 4.1. Human Subjects

Eligible patients were identified by scanning the electronic NF2 patient database at Fondazione IRCCS C. Besta, where the patients had undergone full neurological, ophthalmic, and audiological assessment. Clinical diagnosis was established following the Manchester criteria [51]. All medical records were surveyed and the following data were collected at the time of mutation analysis, and re-verified for accuracy at the time of this study: date of birth, gender, current age, age at diagnosis, age at first symptoms, the time of genetic testing, mode of inheritance, NF2 signs and symptoms including vestibular schwannomas and other cranial schwannomas, spinal and limb schwannomas, meningiomas, NF2 eye features, cutaneous abnormalities, neuropathy, age at first radiotherapy, age at first surgery, and age of starting Bevacizumab. The numbers of schwannomas, meningiomas, and spinal tumors were estimated from brain and spinal Magnetic Resonance Imaging (MRI), with and without gadolinium. Vascular abnormalities were assessed by magnetic resonance angiography of the head.

We also enrolled three patients (two males and one female) with *NF2* intragenic mutations, patients 105, 353, and 359, as controls in RT-qPCR assays. They showed in their *NF2* gene (NM_000268.4) the c.1575-2A>G (p.Lys525Asn fs*209) splice site mutation (patient 105), the c.41_42delCT (p.Leu14Gln fs*34) nonsense mutation (patient 353), and the c.1332dupA (p.Glu445Arg fs*59) nonsense variant (patient 359). Two patients did not have family histories, while in one, the father was affected by NF2 (105). All patients described dizziness and gait disturbance at disease onset, and NF2 was diagnosed at the ages of 40, 14, and 20, respectively. In all cases, brain MRI showed a bilateral VS leading to severe hearing loss. Several brain meningiomas, spinal meningiomas, and schwannomas were also detected in patients 353 and 359. No ocular signs of the disease were observed in any case. In subjects 353 and 359, NF2 plaques were present.

All patients gave their written informed consent to be included in this study as well as for the sampling of their biological material. This study was approved by the ethics committee of the Fondazione IRCCS C. Besta, Milan, Italy (protocol code 4, 11 May 2022).

### 4.2. MLPA Analysis

The SALSA MLPA Probemix P044-C1 NF2 assay (MRC Holland, Amsterdam, The Netherlands) was used for the detection of deletions or duplications in the *NF2* gene in genomic DNA isolated from whole peripheral blood (PB) specimens following the manufacturer’s instructions. The amplification products covered all 17 exons of the *NF2* gene. Product separation was performed using capillary electrophoresis on the ABI Prism 3130 Genetic Analyzer (Thermo Fisher, Waltham, MA, USA) with the GeneScan™ 500 LIZ™ size standard (Applied Biosystems, Foster City, CA, USA). Genemapper Software version 4.0 (Applied Biosystems) was used to analyze sample files collected by the Data Collection Software version 3.0 (Applied Biosystems), and the generated .fsa files were analyzed with the Coffalyser.Net Software (MRC Holland).

### 4.3. qPCR-gDNA

Real-time quantitative PCR on genomic DNA (qPCR-gDNA), extracted from patients’ PB according to standard procedures, was carried out using 50 ng template DNA, according to the standard procedures of the GoTaq–qPCR master mix (Promega, Fitchburg, WI, USA). The sequences of the oligonucleotides and the genomic locations of target regions are shown in Appendix A. Each SYBR Green qPCR assay was run on a QuantStudio 5 Real-Time PCR System (Thermo Fisher). For data analysis, the relative copy numbers of selected regions were determined by comparing the target sequences to the reference (GAPDH), in patient and normal control DNA samples, and applying the formula 2 × 2^(−ΔΔCt)^ [52].

### 4.4. Array CGH Analysis

The array comparative genomic hybridization (aCGH) experiments were performed using Agilent GenetiSure Dx Postnatal Assay Microarray CE-IVD 4x180K+SNP ISCA (International Standards for Cytogenomic Arrays Consortium) with overall median probe spacing of about 25 Kb and 3.5 Kb in ISCA regions (Agilent Technologies Inc., Santa Clara, CA, USA), in accordance with the manufacturer’s instructions. The test DNAs were labeled with CY5-dUT,P and the commercially available sex-matched reference DNAs with CY3-dUTP (Agilent Technologies). The samples were hybridized and washed following standard procedures, and the array was scanned using a SureScan Dx Microarray Scanner CE-IVD G5761AA Agilent at a resolution of 3 μm. The data were extracted and analyzed for copy number changes using Agilent CytoDx v1.2 (ADM-2 algorithm; Agilent Technologies).

### 4.5. Long-Range PCR and PCR

Long-range polymerase chain reactions (LR-PCRs) were performed on patients’ gDNA using Takara La Taq (Takara Clontech, Tokyo, Japan), according to the standard procedures. The oligonucleotides designed for the assays are shown in Appendix A. PCR assays were carried out on patients’ cDNA by means of specific oligonucleotides for the cDNA sequence of AP1B1 (Appendix A), using GoTaq^®^ G2 DNA Polymerase (Promega), following standard procedures.

### 4.6. Sequencing Analysis

The PCR products were sequenced using the Terminator v3.1 Cycle Sequencing Kit (Thermo Fisher) and resolved on an automated ABI-3130xl DNA genetic analyzer (Thermo Fisher). The results were analyzed by means of Chromas Lite software (Technelysium Pty Ltd, South Brisbane, Australia).

### 4.7. Reverse Transcription (RT) and Quantitative Real-Time PCR (qPCR)

Total RNA (500 ng), extracted from patients’ and controls’ PB according to standard procedures, was reverse-transcribed by the Maxima™ H Minus cDNA synthesis master mix with dsDNase (Thermo Fisher). Twelve 22q12 genes reported to be expressed in whole blood in the GTEx portal (https://gtexportal.org, accessed on 8 November 2021) were selected for the study. Four out of twelve genes were mapped in the centromeric region to patient 246’s microdeletion (*HSCB*, *CCDC117*, *XBP1*, and *AP1B1*). The first three genes were also centromeric to the microdeletions in patients 366 and 160. The other eight genes were localized in the telomeric region (*ZMAT5*, *UQCR10*, *ASCC2*, *MTMR3*, *RNF185*, *LIMK2*, *PIK3IP1*, and *PATZ1*). The last seven and four genes were also telomeric to the microdeletions in patients 366 and 160, respectively. The specific oligonucleotides for the qPCR assays are shown in Appendix A. The 2^−ΔCt^ method was applied to determine the gene expression, and the *EIF4A2* gene was used as a housekeeping control for normalization. For each gene analyzed, the mean was calculated in the group of healthy controls, which included ten wild-type subjects, and in a group of three patients with *NF2* intragenic mutations. Each SYBR Green qPCR assay was performed using the GoTaq–qPCR master mix (Promega) and run on a QuantStudio 5 Real-Time PCR Systems (Thermo Fisher). Given the unavailability of biological samples from patients 246, 366, and 160, a formal statistical analysis could not be performed for all the analyzed genes.

### 4.8. Statistical Analysis

Only for the genes found to be dysregulated in the first qPCR assay (*AP1B1*, *MTMR3*, *LIMK2*, and *PIK3IP1*) were biological triplicates carried out. After excluding outliers identified by the Tukey’s test, the independent samples Student’s *t*-test was applied to compare the means, assuming equal variances, and the *p*-values were corrected using the Benjamini–Hochberg (BH) method. The results were considered statistically significant when BH-adjusted *p* < 0.05.

### 4.9. In Silico Analysis

In order to characterize the junction fragments of the microdeletions in patients 246, 366, and 160, we searched for repeated motifs that may have mediated an unequal homologous recombination, resulting in patients’ deletions in the regions containing the BPs, using the Interrupted Rpts Track (Fragments of Interrupted Repeats Joined by RepeatMasker ID) of the UCSC Genome Browser (https://genome.ucsc.edu/, accessed on 8 June 2021). To evaluate the homology between the repeated sequences mapped in the regions containing the centromeric and telomeric BPs, for each patient, we used the Global Alignment Track of the BLAST Browser (https://blast.ncbi.nlm.nih.gov/Blast.cgi, accessed on 8 June 2021).

To study the position effect, we performed predictions of topologically associating domains (TADs) and regulatory elements mapped in the 22q12 region, including the microdeletions in patients 246, 366, and 160 and the flanking regions, by setting the Human GRCh37/hg19 build as the reference assembly. The genomic coordinates chr22:28,250,000–-32,050,000 were chosen for the region to be investigated, according to the position of the first non-deleted TADs, upstream and downstream of patient 160’s microdeletion, which included the deletions of the other two patients. We interrogated the 3D Genome Browser (http://3dgenome.fsm.northwestern.edu/, accessed on 8 November 2021) to view the Hi-C interactions, which mirror TADs, in the GM12878 cell line; VISTA Enhancer Browser (https://enhancer.lbl.gov/, accessed on 8 November 2021) to search for the identified enhancers; and CTCFBSDB 2.0 (http://insulatordb.uthsc.edu/, accessed on 8 November 2021) to identify the predicted insulators. We added to the UCSC Genome Browser two custom tracks, one for VISTA Enhancer, and the other one for the CTCF binding sites, to visualize the regulatory elements included in the region of interest together with the genes.

## 5. Conclusions

Our work thus represents a pilot study for the deep molecular characterization of NF2 patients carrying 22q12 microdeletions and also generally for all other microdeletion syndromes. The characterization of large *NF2* deletions’ BPs allows for identifying their exact gene contents. The identified genes could play pathogenetic roles relevant to specific clinical signs and are also potentially associated with disease severity [13,14]. Furthermore, a deeper characterization of other pathogenetic mechanisms, including the pseudodominance and position effect, could contribute to a better understanding of the pathogenesis of microdeletion syndromes. We also support the routine implementation of microdeletion clinical characterization in NF2 patients within common clinical practice. This strategy will help in addressing the correct estimation of the occurrence of large *NF2* deletions. After MLPA analysis, our patients showed deletions with a maximum extent ranging from 5.9 to 9.6 Mb. Nevertheless, in two of these, deletions were very large, and in one patient, it was sized 147 Kb. For this last patient, microdeletion characterization provided a relevant molecular explanation of the milder phenotype consistency but also showed that the MLPA analysis alone can neither extensively predict the severity of NF2, nor be robust for prognosis. On the contrary, large deletions could easily be used as a prognostic marker of the severe NF2 phenotype, making them useful for patient management to avoid diagnosis delays, thereby facilitating targeted clinical surveillance. Further clinical and genetic studies based on multicenter cohort research, with larger NF2 populations, will not only be fundamental to a better understanding of the NF2 pathogenic basis, but will also provide new insights into the 22q12 genomic architecture and function.

## Figures and Tables

**Figure 1 ijms-23-10017-f001:**
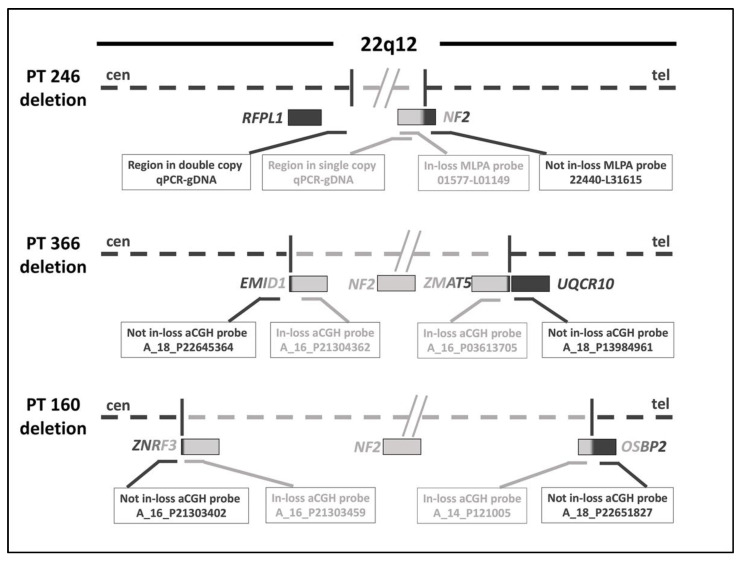
Graphical representation of the 22q12 deleted region in patients 246, 366, and 160. The dotted line in light gray represents the deleted chromosomal region, while the dark line represents the centromeric (cen) and telomeric (tel) non-deleted regions. The filled rectangles represent the genes, colored light gray if deleted, dark gray if not deleted, and shaded if partially deleted, based on the results (shown in the empty rectangles) obtained from the qPCR-gDNA and MLPA assays for patient 246, and by aCGH for patients 366 and 160.

**Figure 2 ijms-23-10017-f002:**
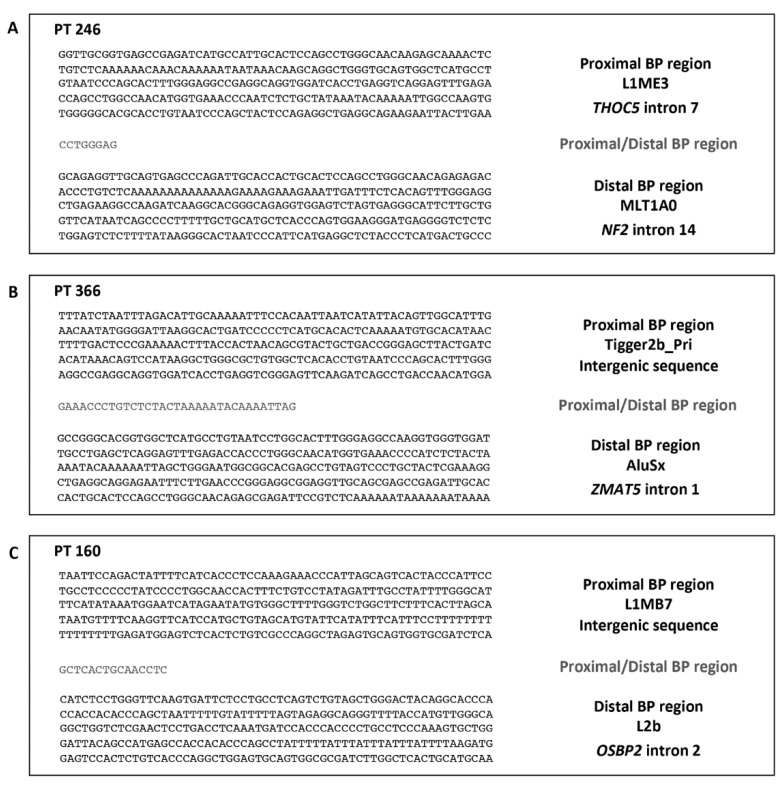
Genomic characterization of the deletion junction fragments. (**A**) Sequence of the deletion junction fragment of patient 246: the proximal BP falls into the L1ME3 repeated element in *THOC5* intron 7, the distal BP falls into the MLT1A0 repeated motif in *NF2* intron 14, and eight base pairs are of uncertain origin, being shared between the two regions. (**B**) Sequence of the deletion junction fragment of patient 366: the proximal BP is in an intergenic region, upstream of *EMID1*, belonging to the Tigger2b_Pri repeat, the distal BP falls into the AluSx repeated motif in the *ZMAT5* intron 1, and thirty-three base pairs are of uncertain origin. (**C**) Sequence of the deletion junction fragment of patient 160: the proximal BP is in an intergenic region, within the L1MB7 repeated element upstream of *ZNRF3*, while the distal BP falls into the L2b repeated motif in *OSBP2* intron 2, while fifteen base pairs are of uncertain origin.

**Figure 3 ijms-23-10017-f003:**
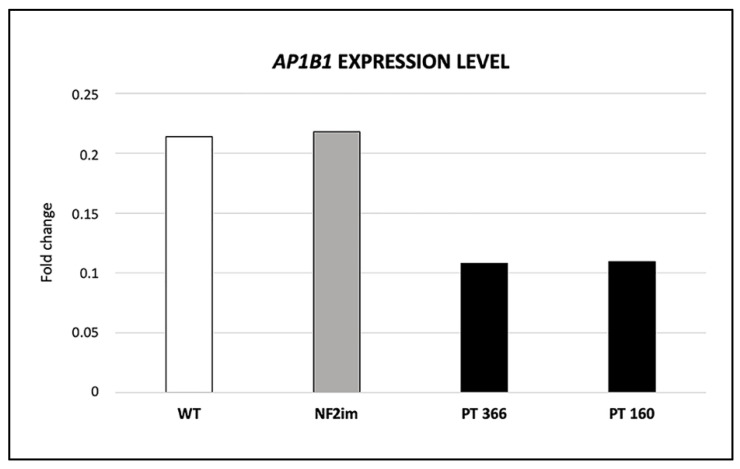
Expression level of the *AP1B1* gene. The *AP1B1* quantitative expression level (2^−ΔCt^) in the peripheral blood of patients 366 and 160 has been compared to the average value of ten healthy controls (WT) and three patients with *NF2* intragenic mutations (NF2im). The *AP1B1* mRNA was expressed at approximately half in our patients.

**Figure 4 ijms-23-10017-f004:**
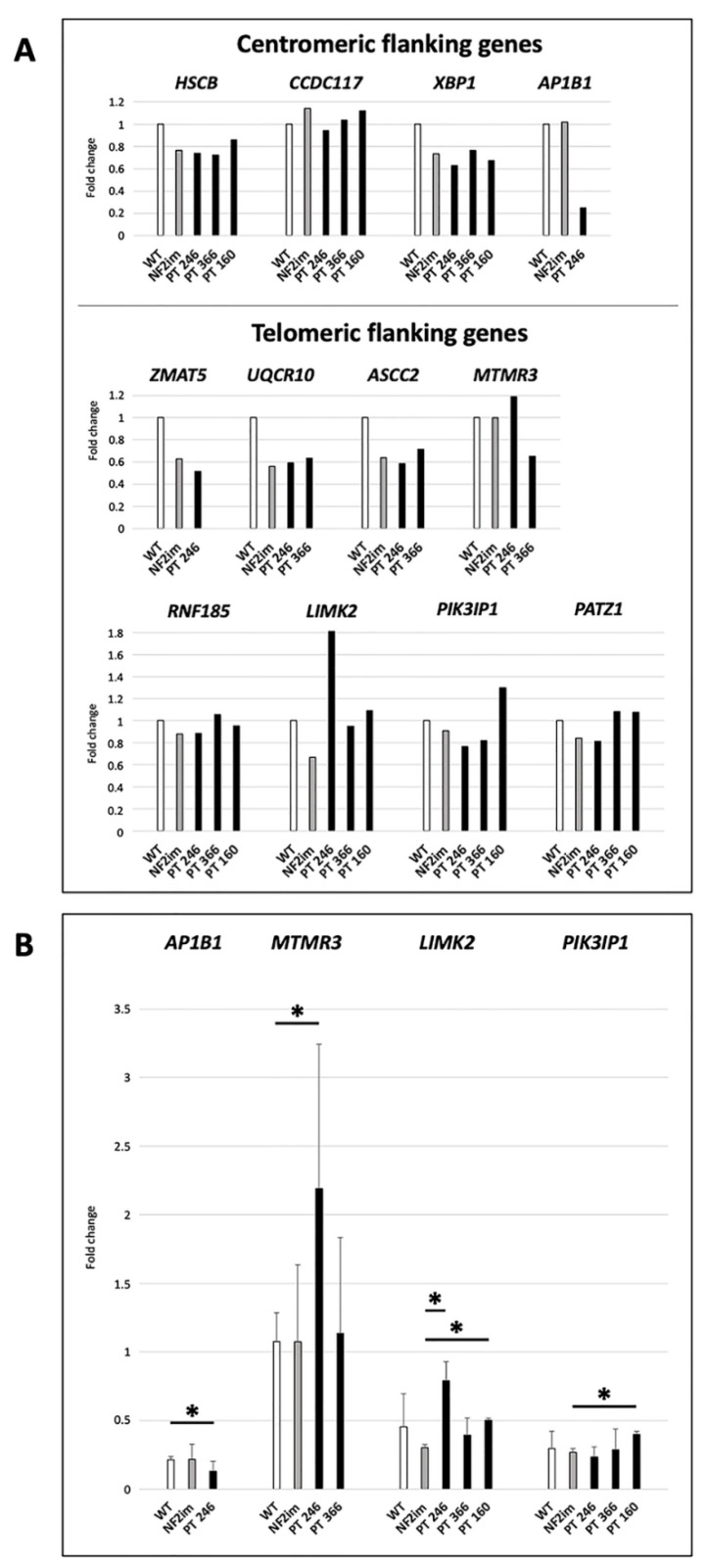
Gene expression analysis of the selected genes flanking the patients’ deletions. (**A**) The histogram shows quantitative expression levels (2^−ΔCt^) in the peripheral blood of 4 centromeric and 8 telomeric genes to the patients’ deletions, in our 3 NF2 microdeletion patients compared to the mean of 10 wild-type subjects (WT) and to the mean of 3 NF2 patients with intragenic mutations (NF2im). For comparison, the mean WT expression level was set as equal to 1. Among the selected genes, *AP1P1*, located centromerically to patient 246’s deletion, was deleted in patients 366 and 160, and between telomeric genes, *ZMAT5* was removed from the deletions of patients 366 and 160, while *UQCR10*, *ASCC2*, and *MTMR3* were included in patient 160’s deletion. A trend towards dysregulation was found for *LIMK2* in patient 246, *MTMR3* in patients 246 and 366, *LIMK2* in patients 246 and 160, and *PIK3IP1* in patient 160 only, by comparison with both wild-type controls and NF2 patients with intragenic mutations. (**B**) The histogram shows the quantitative expression levels, in peripheral blood, of the previously identified dysregulated genes. For each patient, the value is mean ± standard deviation (SD) from three independent biological samples, while for the ten healthy controls and three *NF2* intragenic mutation patients, the values are means ± SD. * BH-adjusted *p* < 0.05, Student’s *t*-test.

**Figure 5 ijms-23-10017-f005:**
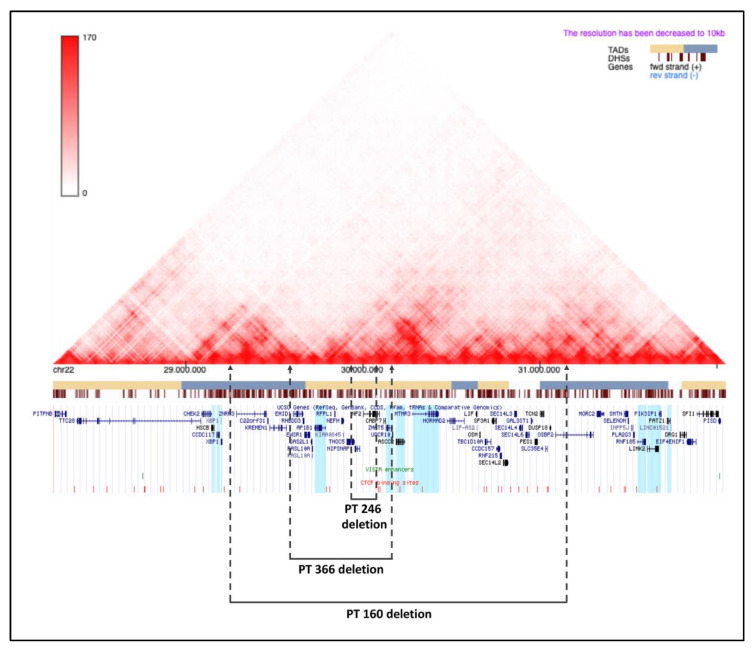
Visualization of topological domains mapping within the 22q12 genomic region. The heatmap, obtained from Hi-C data from blood-derived GM12878 lymphoblastoid cells, shows 22q12 DNA–DNA interactions and inherent TADs, whose positions along the chromosome are indicated by the yellow and cerulean blue bars. The barcode in brown represents the DNase I hypersensitive sites (DHSs). The UCSC genome browser screen shows the genes (in blue and black; those whose expression level has been evaluated by qPCR are highlighted in light blue), VISTA enhancers (in green), and CTCF binding sites (in red). The dashed lines correspond to the patients’ deletion breakpoints. Patient 246’s deletion, encompassing a single TAD, removed three genes, but no regulatory elements. Patient 366’s deletion partially involved two TADs, including 13 protein-coding genes and four CTCF-binding sites. Patient 160’s deletion, encompassing five TADs, included 38 protein-coding genes and 18 CTCF binding sites. Two enhancers were mapped outside the patients’ deletions.

**Table 1 ijms-23-10017-t001:** Clinical features of NF2 microdeletion patients.

Patient		246	366	160
Gender		Female	Male	Male
Sporadic/familial		Familial	Sporadic	Sporadic
Age at first neurological symptom		23	7	3
Age at diagnosis		27	13	22
Current age		29	44	24
Tumor load	Bilateral VS	Yes	Yes	Yes
	Unilateral VS	/	/	/
	Intracranial meningioma	No	No	No
	Non-VS intracranial schwannoma	No	Yes	Yes
	Spinal meningioma	No	No	No
	Spinal schwannoma	No	Yes	Yes
	Spinal ependymoma	No	No	No
Ocular features	Epiretinal membranes	No	No	No
	Cataract	No	Yes	No
	Combined hamartoma	No	No	No
	Optic nerve meningioma	No	No	No
Hearing loss	Bilateral severe hearing loss	Yes	No	Yes
	Total hearing loss in at least 1 ear	No	Yes	No
Neurological features	Cranial nerve palsy VI, VII lower CNP	No	Yes	Yes
	Seizures	No	No	No
	Gait disturbance	Yes	Yes	Yes
	Mental retardation	No	Yes	Yes
Dermatological features	Hypopigmentation	No	No	No
	Hyperpigmentation	Yes	Yes	Yes
	Skin lumps	No	Yes	Yes
	NF2 plaques	No	Yes	No
	Schwannoma	No	No	Yes
Vascular abnormalities		No	No	Yes
Neuroradiological anomalies	Cerebellar hypoplasia	No	No	No
	Corpus callosum hypoplasia	No	No	Yes
	Prominent perivascular spaces with white matter loss	No	No	Yes
	Prominent choroid plexus	No	No	No
	Focal cortical dysplasia	No	No	No
	Cerebellar hamartoma	No	No	No
Other	Mental disability	No	Yes	Yes
	ADHD	No	Yes	No
	Scoliosis	No	No	No
Interventions	VS surgery	No	Yes	No
	Non VS intracranial surgery	No	No	No
	Spinal surgery	No	No	No
	Shunt surgery	No	Yes	No
	Radiotherapy	Yes (two times)	Yes	No
	Bevacizumab	No	No	Yes
Age at first radiotherapy session		27	30	/
Age started bevacizumab		/	/	23
Age at first surgery		/	30	/

Yes, present; No, absent; /, not applicable; VS, vestibular schwannoma; ADHD, attention deficit hyperactivity disorder.

## Data Availability

The data about the clinical signs of the three patients and the extents of their deletion will be stored in the LOVD repository (https://www.lovd.nl, accessed on 30 August 2022).

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
