# Peer review of "Characterization of 22q12 Microdeletions Causing Position Effect in Rare NF2 Patients with Complex Phenotypes"

_ijms, 2022, doi:10.3390/ijms231710017_

Round 1
Reviewer 1 Report
The authors here report a comprehensive characterization of breakpoints in NF2 patients with microdeletions differently affecting the gene, exploring by in silico approaches and expression analysis how the loss of TADs and regulatory elements can have an impact of the observed variability from mildest to more severe phenotypes. The manuscript is interesting, and it moves in a little explored land for the neurofibromatosis type 2. However, the main limitation is represented by the very small number of NF2 patients investigated, and it would require the characterization of additional cases.
Main criticisms:
1. Pt 246 is a familial case but no information is provided on clinical features of the other two affected relatives. Do they have very similar manifestations or a very different clinical presentation? In an attempt to correlate microdeletion features to the phenotype, this appears to me an important point to clarify.
2. For evaluating positional effect of NF2 microdeletions on flanking genes, authors compare expression level of selected genes in blood samples from the three patients studied, healthy controls, and three other patients with NF2 intragenic mutation. For the last group, no information is provided on the type of mutations and their effect on NF2 expression. Is it similar for all NF2 patients they compared? This point should be clarified.
3. In Discussion section, the authors suggest that haploinsufficiency of AP1B1 can justify ID in Pt 366 and 166. In my opinion, there are not sufficient evidence supporting this, and the sentence at lines 458-461 should be rephrased.
Minor criticisms:
1. Figure 2 seems uninformative and could be removed. Alternatively, the repeated elements mediating rearrangements could be highlighted.
2. line 305: correct heathy as healthy.
Author Response
Reviewer #1 (Comments to the Author):
Main criticisms:
- Pt 246 is a familial case but no information is provided on clinical features of the other two affected relatives. Do they have very similar manifestations or a very different clinical presentation? In an attempt to correlate microdeletion features to the phenotype, this appears to me an important point to clarify.
We checked father’s an aunt’s clinical files and available clinical information were added (page 4, lines 127-130).
- For evaluating positional effect of NF2 microdeletions on flanking genes, authors compare expression level of selected genes in blood samples from the three patients studied, healthy controls, and three other patients with NF2 intragenic mutation. For the last group, no information is provided on the type of mutations and their effect on NF2 expression. Is it similar for all NF2 patients they compared? This point should be clarified.
Information regarding the type of mutation and the clinical phenotype of patients with NF2 intragenic mutation has been added (page 16, lines 546-556).
- In Discussion section, the authors suggest that haploinsufficiency of AP1B1 can justify ID in Pt 366 and 166. In my opinion, there are not sufficient evidence supporting this, and the sentence at lines 458-461 should be rephrased.
The sentence has been rephrased (page 15, lines 470-475).
Minor criticisms:
- Figure 2 seems uninformative and could be removed. Alternatively, the repeated elements mediating rearrangements could be highlighted.
We modified Figure 2 by adding the name of the repeated elements that mediate the rearrangements (page 8).
- Line 305: correct heathy as healthy.
The typo has been corrected (page 10, line 329).
Please see the attachment.

Reviewer 2 Report
Genetic features have not been precisely characterized in NF2 patients with 22q12 microdeletion. In this manuscript, Tritto et al. used a panel of genomic techniques to identify and characterize the extend of microdeletion in correlation with the clinical description of three NF2 patients. NGS and expression analyses allowed to precisely identify breaking points, gene content, position effects on neighboring genes. This type of detailed analysis can provide a better genotype-phenotype correlation and prognostic markers for the NF2 patients.
The manuscript is easy to read and method/results well described. The Discussion, although quite long, includes very interesting considerations regarding consequences of microdeletion for the pathogenesis, care and diagnosis of these patients.
Main comments:
Intro: Lines 92-94: sentence not clear.
Results: Lines 251-255: The authors might rephrase to avoid repetition (involvement of 5’ portion) and double check the accuracy of lines 252-253: “chimeric gene was expected”?
Results: Lines 256-269: The authors should include and explain the possibility or not for fusion genes in patients 366 and 160.
Discussion: Lines: 447-461 are describing new results and a new figure so should be move to the Results section.
Figure 3. Panel 3a: missing error bars. It is not clear why the authors created a second panel 3b with the same data.
Minor comment:
Now days, medical English uses "T/thoracic" for designation of the thoracic vertebrae.
Author Response
Reviewer #2 (Comments to the Author):
Main comments:
- Intro: Lines 92-94: sentence not clear.
The sentence has been rephrased (page 2, line 92-94).
- Results: Lines 251-255: The authors might rephrase to avoid repetition (involvement of 5’ portion) and double check the accuracy of lines 252-253: “chimeric gene was expected”?
The sentence has been rephrased (page 7, lines 246-249).
- Results: Lines 256-269: The authors should include and explain the possibility or not for fusion genes in patients 366 and 160.
This information has been provided (pages 7-8, lines 250-262).
- Discussion: Lines: 447-461 are describing new results and a new figure so should be move to the Results section.
A paragraph describing the mutation and expression analysis of AP1B1 has been included in the Results section (page 9, lines 276-297). Accordingly, the Figure S7 is now the Figure 3 (page 9).
- Figure 3. Panel 3a: missing error bars. It is not clear why the authors created a second panel 3b with the same data.
The Figure 3 is now the Figure 4. As explained in the Materials and Methods section (4.7-4.8 Subsections, page 18, lines 621-626), for the unavailability of biological samples from our patients, we could not carry out a formal statistical analysis for all the analyzed genes, but only for the few genes for which it was possible to perform the analysis in triplicate, shown in panel 4b.
Minor comment:
- Now days, medical English uses "T/thoracic" for designation of the thoracic vertebrae.
The shortcut has been corrected (page 5, line 149).
Please see the attachment.
Round 2
Reviewer 1 Report
I thank the authors very much for having considered my concerns, modifying the manuscript in accordance. Remain the limitation represented by the very small number of NF2 patients investigated. I'm confident authors will extend the analysis to additional cases in the next future.
Author Response
We agree that the very small number of NF2 patients investigated represented a limitation of the paper and we address the point in the conclusions page 19 lines 684-687. In the next future wes will extend the analysis to additional cases an
As suggested, we submitted the manuscript to MDPI for English editing.